# A Multisensory Network Drives Nuclear Mechanoadaptation

**DOI:** 10.3390/biom12030404

**Published:** 2022-03-04

**Authors:** Asier Echarri

**Affiliations:** Centro Nacional de Investigaciones Cardiovasculares (CNIC), Mechanoadaptation and Caveolae Biology Laboratory, Areas of Cell & Developmental Biology, Calle Melchor Fernández Almagro, 3, 28029 Madrid, Spain; aecharri@cnic.es

**Keywords:** nucleus, mechano-transduction, nuclear envelope, lipid bilayer mechano-sensing, mechanosensitive molecules

## Abstract

Cells have adapted to mechanical forces early in evolution and have developed multiple mechanisms ensuring sensing of, and adaptation to, the diversity of forces operating outside and within organisms. The nucleus must necessarily adapt to all types of mechanical signals, as its functions are essential for virtually all cell processes, many of which are tuned by mechanical cues. To sense forces, the nucleus is physically connected with the cytoskeleton, which senses and transmits forces generated outside and inside the cell. The nuclear LINC complex bridges the cytoskeleton and the nuclear lamina to transmit mechanical information up to the chromatin. This system creates a force-sensing macromolecular complex that, however, is not sufficient to regulate all nuclear mechanoadaptation processes. Within the nucleus, additional mechanosensitive structures, including the nuclear envelope and the nuclear pore complex, function to regulate nuclear mechanoadaptation. Similarly, extra nuclear mechanosensitive systems based on plasma membrane dynamics, mechanotransduce information to the nucleus. Thus, the nucleus has the intrinsic structural components needed to receive and interpret mechanical inputs, but also rely on extra nuclear mechano-sensors that activate nuclear regulators in response to force. Thus, a network of mechanosensitive cell structures ensures that the nucleus has a tunable response to mechanical cues.

## 1. Introduction

Adaptation of cells to environmental changes is essential for homeostasis, cell survival, and successful proliferation. This adaptation necessarily implies sensing the environmental variables, such as temperature, chemical diversity, and mechanical cues, which impact on cell functions [1]. Once these cues are sensed, they must be transduced into structural and chemical changes throughout the cell, including the nucleus [2]. Mechanical cues are inherent to life, and gravitational forces represent a clear example of the universality of mechanical forces acting on living organisms. In addition to this universal force, at microscopic level, cells experience multiple types of mechanical forces that originate outside cells, such as shear stress, stretching and compression, and within cells, such as osmotic swelling or actin cytoskeleton/myosin II-driven contraction [3]. Furthermore, the diverse environmental architecture must be interpreted by the cell and translated into nuclear function [4].

The process of sensing mechanical cues, or mechanosensing, is followed by a process of translating this information into chemical modifications, known as mechanotransduction. Evolution has selected molecules and macromolecular complexes that have the capability to sense forces [5]. To date, only a handful of molecules have been well studied with respect to their mechanosensing properties [5,6], but this has allowed us to begin to understand the molecular basis of mechanotransduction pathways.

The prototypic mechano-sensing molecules are associated with focal adhesion complexes and the actin cytoskeleton. These mechanosensitive molecules modify their conformation, ligand affinity and/or binding partners as a result of force application [7]. This is the case for the extracellular matrix (ECM) component fibronectin, integrins, and adaptor molecules located at focal adhesions, such as talin or filamin A [7,8]. These mechanosensors create a molecular chain, and each of the units, in this case mechanosensitive molecules, are modified by force in a chain reaction fashion, which triggers downstream signaling cascades, resulting in a cell mechanoresponse and mechanoadaptation [6,7].

Mechanistically, the machinery forming focal adhesions operates with forces generated at: (i) the interface ECM–integrins, and (ii) by the polymerization/contraction of actin filaments, to which talin and other actin binding molecules are bound (Figure 1). Thus, pulling forces generated by the actomyosin are believed to unfold talin domains, exposing new binding sites for vinculin [9] (Figure 1). It is believed that tyrosine kinases at focal adhesions operate by similar principles [10,11,12]. As a result, these and additional force-driven modifications initiate mechanotransduction. Using this rationale, any molecule bound to the actin filaments and to another structure could potentially sense the forces generated by the actomyosin system. This basic mechanism, which translates mechanical information up to the nucleus by mechanisms detailed below, is likely to operate in many cellular locations, but evidence in this regard is scarce. The scheme based on the ECM-integrins-adaptors-actin can be highly flexible and sensitive (Figure 1) [13], and allows the sensing a range of cues; however, this mechanism could hardly transduce the full range of mechanical cues existing in biological systems.

In parallel to this general scheme, based on adhesion receptors and the actin cytoskeleton (Figure 1), another scheme is emerging (Figure 2), where force-driven structural alterations in lipid bilayers, at lipid packing, membrane curvature degree, or vesicle trafficking, trigger a response in proteins functionally and/or physically associated with those lipid domains. These protein responses initiate signaling events that contribute to mechanoadaptation. As detailed below, these mechanisms are especially relevant for mechanotransduction to the nucleus. An important feature of this mechanism of transferring mechanical information is its independence on adhesion complexes, and some of these lipid domain mechanoadaptations are also independent on the forces generated by the actomyosin system [14] (Figure 2).

Since the initial observations provided by the Ingber lab suggesting that nuclei were hard-wired with the cell surface [15], numerous studies have identified the molecular architecture that connects the extracellular matrix to the nucleus [2,4]. In this review, I provide an overview of the physical connections that exist between the cell surface and the nucleus, and the multiple mechanisms that are in place to ensure that the nucleus perceives mechanical cues. The current understanding suggest that cells have a multisensory network that senses and transduces a plethora of mechanical cues to ensure a tailored nuclear mechanoresponse.

## 2. Structural Elements of the Nucleus Relevant for Mechanosensing and Mechanotransduction

The nucleus has several distinctive structural characteristics that make it a unique and highly specialized organelle. Several of its exclusive structures, including the nuclear envelope (NE), the nuclear pore complexes (NPCs), the linkers of the cytoskeleton and nucleoskeleton (LINC), the lamina and the chromatin, have physical properties compatible with being mechanosensitive structures. Although in many cases we still lack in vitro information demonstrating that they are indeed mechanosensitive at the molecular level, the current knowledge suggests that their function and behavior is mechanoresponsive and/or mechanosensitive. Thus, the nucleus, as a single unit, acts as a mechanosensory organelle [16,17,18] that mechanoadapts gene expression and DNA protecting pathways [19,20,21], and even functions associated with the whole cell, such as migration [17,18]. Hence, the nuclear molecular composition must be able to sense, adapt and transduce mechanical forces that are sensed by the cell surface, and by the nucleus itself (Figure 2 and Figure 3).

### 2.1. The Nuclear Envelope: A Perforated Barrier for Mechanosensing

The nuclear envelope is formed by two lipid bilayers, the outer nuclear membrane (ONM) and the inner nuclear membrane (INM), that define the limits of the nuclear envelope lumen or perinuclear space (PNS) (Figure 3). The NE is continuous with the endoplasmic reticulum (ER) and is considered to be a specialized form of ER that has its own protein composition and structural particularities [22,23]. The NE can be divided into three different regions based on its curvature: (i) Regions where the curvature of the ONM and INM is abruptly increased, resulting in the fusion of both membranes to create a pore, in which the nuclear pore complex is inserted [24]; (ii) regions that also sharply bend and form invaginations towards the nuclear interior and that present different shapes and sizes. Whether all these invaginations, known as nuclear grooves [25], nucleoplasmic reticulum [26] or nuclear branches [27], correspond to the same functional structure is currently unclear; (iii) smoothly curved convex regions, that occupy most of the nuclear envelope, and shape the nucleus as we know it.

The nuclear envelope plays an important role in the nuclear mechanoresponse. Some of the nuclear invaginations or indentations have been shown to be reduced by cell stretching, suggesting that they are mechanosensitive structures [28]. Some of these deformations are dependent on the actin cytoskeleton and lamin B1 [25,27,29], again suggesting that they may be important in the mechanoresponse of the NE. Indeed, two recent studies have associated the unfolding of the NE in response to compression with a cell mechanoresponse to evasion in tight 3D environments [17,18]. On the contrary, in skin epidermis stem/progenitor cells, short duration stretching induces an increase in NE wrinkles correlating with NE softening [30], highlighting the plasticity of the NE. Although little is known regarding the pathways controlling the local curvature of the NE, a pathway dependent on ATR, a mechanoresponsive kinase activated by genotoxic stresses, maintains the smoothness of the NE [31,32]. The plasticity of the NE is also observed when cells transverse tight spaces, producing significant mechanical stress. Local NE curvature is severely induced by the formation of nuclear protruding blebs when nuclei must traverse tight spaces, which may lead to NE rupture and DNA damage [20,21,33].

The nuclear envelope, like the plasma membrane, also regulates signaling events. Swelling, compression or stretching of the NE has been proposed to alter the organization of its lipids [34], which would alter the affinity of certain enzymes. This process is likely to regulate the association of cPLA_2_ (cytosolic phospholipase A_2_, which catalyzes the hydrolysis of the sn-2 position of membrane glycerophospholipids to release free fatty acids) with the INM. Upon cell swelling induced by hypo-osmotic shock, or nuclear compression, cPLA_2_ moves from the nucleoplasm to the INM, triggering its activation, a process that is counteracted by polymerization of actin filaments and Lamin A/C [35]. The interpretation of these results is that tension increase at the INM by osmotic swelling or compression, produces alterations in lipid packing, which favors the affinity of cPLA_2_ for its INM binding sites [34,36]. In accordance with this model, experiments in cell-free systems have shown that binding of the C2 domain of cPLA_2_ to artificial lipid bilayers is modulated by stretching, suggesting that this domain is mechanosensitive [37]. Interestingly, amphipathic lipid packing sensor (ALPS) motifs, which sense bent regions of the membrane due to its capability to recognize defects in lipid packing that arise from membrane deformation [38], competes with the cPLA_2_ C2 domain for membrane binding, suggesting that both recognize similar binding sites on curved membranes [37] (Figure 4a). This mechanism, mediated by calcium, produces arachidonic acid, which triggers a signaling response important for leukocyte chemotaxis in damaged tissues [39], and myosin II-mediated contraction for cell evasion in tight 3D environments [17,18].

Together, these studies suggest that the NE is mechanosensitive, is coupled to mechanosensitive molecules, and is plastic with respect to its response to mechanical stimuli (Figure 4a,b). In addition to the lipid bilayers of the NE, there are two exclusive components within the NE that play a major role in nuclear mechanoresponse: the LINC complex and the NPC (see next).

### 2.2. LINC Complexes Act as the Antennas of the Nucleus

The LINC complex is composed of KASH (Klarsicht, ANC-1, Syne homology) proteins that locate to the ONM, and SUN (Sad1 Unc-84) proteins that locate in the INM. KASH proteins are also known as nesprins (nuclear envelope spectrin repeat, 1 to 4) [44], and in mice were originally named syne, for synaptic nuclear envelope [45]. This family will be referred to as nesprins. Nesprins contain a KASH domain in the C-terminus that expands through the ONM and interacts with SUN proteins in the PNS [40,46,47]. Nesprins protrude towards the cytoplasm where they directly or indirectly interact with the cytoskeleton; as a result, a connection between SUN proteins and the cytoskeleton is built. Potentially, some of the nesprin isoform can reach long distances in the cytosol due to their size (nesprin1 isoform 1, or giant (1G), has 1011 kDa, and is predicted to contain up to 74 spectrin repeat (SR) domains); for this reason, they can be visualized as antennas allowing the nucleus to connect with the cytosol (Figure 3) [48]. Nesprin 1 and 2 interact with the actin cytoskeleton through two paired calponin homology (CH) domains in their N-terminal region, and indirectly with microtubules through motor proteins binding to the SR domains, as nesprin 4 does [40]. Nesprins also interact with additional proteins; in the case of nesprin 2G, it interacts with formin FHOD1, reinforcing its linkage with the actin cytoskeleton [49]. Finally, connections with intermediate filaments, bridged by plectin, have been described for nesprin 3α [50]. Interestingly, connections between the actin cytoskeleton and nesprin 3α also exists. Plectin contains an actin binding domain (ABD) and nesprin 3α interacts with the ABD of nesprin 1G and 2 [51]. In addition, nesprins bind many other molecules that are not necessarily related to the cytoskeleton [52,53].

The connections with the nuclear interior are mediated by SUN 1 and 2 proteins. SUN proteins bind several proteins inside the nucleus, including lamins [47,54], Samp1 [55] and emerin [56]. This allows the creation of a chain from the cytoskeleton, outside the nucleus, up to the nuclear lamina (Figure 3). Accordingly, mechanical information gathered and generated by the cytoskeleton can be transduced through the LINC complex to the lamina and the chromatin (see below) [48] (Figure 4b).

Multiple studies have shown that the LINC complex is required for efficient nuclear mechanotransduction [57], nuclear positioning and movement [40]. A seminal study by Lammerding et al. showed that mechanically induced nuclear deformations are dependent on the LINC complex [58], identifying key molecular components mediating initial observations [15]. Similarly, Nesprin3 is required for shear forces driven endothelial cell migration and centrosome positioning [59]. In addition, Nesprin2g and 3 are required for shear induced actin cap reorganization [60]. Additional evidence indicating that nesprins mediate the mechanoresponse of the nucleus was obtained by mechanically stimulating isolated nuclei [16]. This approach showed that nesprin1 mediates the mechanoresponse of the nucleus to mechanical stimulation and that this pathway depends on SUN, Lamin A/C and emerin [16].

Some evidence suggests that nesprins are mechanosensitive molecules and unfold in response to mechanical stretching [61,62] (Figure 4b). Mini Nesprin2G has been used to generate a Förster resonance energy transfer (FRET)-based tension biosensor, and experiments directed to validate its function suggest that nesprins change their architecture as a function of tension in the cell. Whether this is a direct effect of pulling forces is not fully determined, but other studies suggest that this is likely. Spectrin-like repeats of nesprin1α have physico-chemical properties similar to spectrins [63], and in vitro spectrin’s SR domains undergo stepwise stretching and unfolding when subjected to pulling forces [64]. Furthermore, the characterization of the crystal structure of the partial LINC complex suggested that the complex is suitable for its mechanical role in transmitting cytoskeleton generated forces [65,66]. Taken together, these studies suggest that nesprins are likely to be mechanosensitive molecules, but additional direct evidence is needed to understand how they operate exactly.

### 2.3. Nucleocytoplasmic Shuttling and the Nuclear Pore Complex Are Mechanically Regulated

The NPC is the tunnel through which to enter the nucleus. While ions and small molecules freely diffuse through it, molecules larger than ~5 nm or ~40 KDa require assistance by a nuclear transport receptor (NTR) to cross it [67]. The NPC functions as a permeability barrier composed of phenylalanine-glycine (FG) repeat domains present in some nucleoporins, which are the proteins forming the NPC [67]. There are ~2–3000 NPC units per average eukaryotic cell [67] and some evidence suggests that differences in the protein stoichiometry may occur between different cell types [68]. Recently, this concept has been reinforced by Schwartz et al., showing that the diameter of the NPC is modulated by the cellular environment [41]. Little is known about the sensitivity or response of the NPC to mechanical cues, but the fact that differences in the NPC diameter are observed as a function of ECM rigidity suggest that mechanical cues may impact in the composition and/or structure of the NPC [42]. Indeed, a recent study has shown that starving and hyper-osmotic shock significantly alters the diameter of the NPC [43]. Taken together, these studies suggest that the NPC is relatively plastic [69] and is likely to function as a mechanoresponsive macromolecular structure (Figure 4c).

In addition to the NPC, the shuttling system driving nuclear proteins inside the nucleus is also modulated by mechanical cues. A recent study has shown that the NTR Importin 7 (Imp7), which imports cargoes into the nucleus [70], is highly mechanoresponsive. Upon increased tension in the cell, Imp7 is localized to the nucleus, while it is cytoplasmic under low tension conditions [71]. Interestingly, Garcia-Garcia et al. showed that Imp7 is the NTR responsible for translocating into the nucleus the main transcriptional regulator downstream of mechanical cues, named YAP. As a consequence, Imp7 is indispensable for YAP-induced organ growth in vivo [71]. The interaction between YAP and Imp7 also regulates nuclear import. When mechanical cues activate YAP, it binds Imp7, which controls its mechanoresponse. In addition, this complex restricts the binding to Imp7 of other of its cargoes, such as Smad3 and Erk2, limiting their nuclear localization [71]. Therefore, mechanical cues, through the Hippo pathway and YAP, control the nuclear import pathway led by Imp7, downregulating other pathways dependent on this entry route [71]. Thus, not only the NPC, but the regulators of the nucleo-cytoplasmic shuttling are regulated by mechanical cues.

### 2.4. The Nuclear Lamina Is Essential for Nuclear Mechanical Stability and Mechanoresponse

Below the NE, a ~10–30 nm wide meshwork provides mechanical and structural stability to the nucleus; this structure is known as the lamina and plays a major role in cell mechanobiology [72]. The major constituent of the nuclear lamina is a filamentous network composed of two types of proteins, A-type and B-type lamins: A-type lamins are encoded by *LMNA* that gives rise to Lamin A, Lamin C -by alternative splicing- and other rare isoforms, AΔ10 and C2. B-type lamins include B1 (encoded by *LMNB1* gene), B2 and B3 (encoded by *LMNB2* gene) [73]. Lamins belong to the type V intermediate filaments family and as such they form filamentous structures [74,75]. Lamin A, C and B-type lamins, despite their sequence and structural similarities [76], form distinct filamentous networks within lamina [77]. Lamins assemble into two long-coiled coil dimers that form a tetramer, giving rise to a ~3.5 nm thick filaments, making them quite different from other cell filamentous structures, which are much wider [74]. Lamin filaments, with an average length of 380 ± 122 nm, show a high degree of flexibility [74]. The mechanical properties of intermediate filaments have been probed using various approaches. These studies have shown that they are quite flexible structures with viscoelastic properties [78].

There is evidence suggesting that the localization of the lamin B1 meshwork, which is closer to the INM than Lamin A/C (Figure 3), is curvature and strain-responsive [79], although it does not contribute much to nuclear stiffness [80,81]. The localization of Lamin A/C is also susceptible to regulation. Lamin A/C is not evenly distributed underneath the NE and modifications in its organization are inferred based on the differential staining of lamin A/C in the apical region of the nucleus compared to the basal region. These differences are modulated by substrate rigidity, compression forces, and are dependent on an intact LINC complex and the perinuclear actin filaments [82]. These results suggest that mechanical cues somehow are modulating the way lamin A/C is organized below the NE. Similarly, a phosphorylation in lamin A/C have been shown to be associated with changes in substrate rigidity [83]. Confirming that Lamin A may be mechanosensitive, cysteine shotgun mass spectrometry-based analysis showed that cysteine 522 was sensitive to mechanical stress [84] (Figure 4b). These studies suggest that lamin A/C is capable of mechanosensing, as such, it is an essential structure to transmit force to the chromatin [85]. Indeed, force application in isolated nuclei induces a recruitment of Lamin A/C to nesprin1-associated complexes [16]. As expected, lamins play a major role in defining the mechanical properties of stiff and viscoelastic nuclei [78,86,87], and in transmitting mechanical force to the nucleus to regulate gene expression [81,88,89,90]. As detailed below, the nuclear lamina is connected with the chromatin, which ensures transmission of force to gene expression.

### 2.5. The Chromatin Is Bound to the Mechanosensitive Nuclear Elements

The chromatin, composed of histones and DNA, is organized into structures with increasing complexity, from small nucleosomes, to chromatin fibers, and up to chromosome territories [91]. This organization in levels of complexity is susceptible to dynamic changes in the genome, which is coupled to regulation of DNA transcription, replication and repair [91]. Histone post-translational modifications permit remodeling of the chromatin to allow gene expression and access for transcription factors [92,93]. Chromatin is divided into two types depending on its “activation” status. Heterochromatin is tightly packed and not active for transcription; on the contrary, euchromatin is loosely packed and transcription is more active. Histone post-translational modifications play a major role in the organization of chromatin domains and several studies have shown that histone post-translational modifications (PTMs) are regulated by mechanical forces and that they are important for nuclear stiffness [94,95,96,97,98,99,100,101]. Indeed, histone 3 methylation (H3K9me3) reduction and other PTMs are essential to induce nuclear softening in response to acute mechanical stretching [30]. How exactly histone PTMs are controlled by mechanical stimuli is not clear, but nucleocytoplasmic shuttling, sensitive to mechanical cues, has been described for HDAC3 [102].

Chromatin regulation is determined partially by its association with mechanosensitive lamina, in regions known as lamina associated domains (LADs), which are enriched in silenced heterochromatin [103]. The linkage between chromatin and the lamina is important for regulation of gene expression and cell differentiation [104]. Several proteins are involved in linking both mechanoresponsive structures. LEM (LAP2-emerin-MAN1) domain proteins, such as emerin, LEMD2 and LEMD3/MAN1, bind lamins and chromatin [105], providing a molecular link between lamina and chromatin (Figure 3). This family binds barrier-to-autointegration factor (BAF), a histone- and DNA-binding protein, acting as a bridge to the chromatin [105]. There is evidence that LADs are sensitive to mechanical strain [95], and mechanoresponsive emerin [16] plays a role in this process [95].

The fact that chromatin is indirectly connected with the lamina suggests that it could respond to mechanical signals. Direct evidence of this model came from Wang et al. Mechanical stimulation of cells through integrins induced a rapid rearrangement of the chromatin, resulting in upregulation in the expression of a reporter transgene of dihydrofolate reductase (DHFR) [19]. This force-induced chromatin deformation is dependent on SUN, lamins, BAF, HR1 and emerin [19], and H3K9me3 demethylation [106]. Thus, mechanical cues regulate chromatin remodeling to control gene expression [19,106] and DNA protection [30]; in turn this chromatin remodeling affects the mechanical properties of the nucleus [30,94]. Together these studies show that chromatin is an active player in the nuclear adaptation to mechanical cues.

### 2.6. The Cytoskeleton Transmits Force to the Nucleus

The cytoskeleton generates forces that play a major role in mechanotransduction [107]. The actin cytoskeleton and myosin II complex, known as the actomyosin system, form the core of the stress fibers [107,108]. In addition, stress fibers contain additional actin binding proteins that are required for their architecture and function [107,108]. Stress fibers play a major role in the cell response to mechanical cues [107], as they are physically connected with focal adhesion proteins [107]. Polymerization and contraction processes generate the forces that are transduced into biochemical and/or physical changes in actin binding proteins, which initiate diverse signaling cascades (reviewed in [7,107]) (Figure 1). In the cytosol, this process regulates multiple cytosolic factors involved in mechano-transduction, but cytosolic stress fibers also regulate mechanotransduction directed to the nucleus, directly and indirectly [109,110,111,112].

A direct mechanism to transduce mechanical forces sensed at focal adhesions is mediated by a set of stress fibers that form over the nucleus, connecting both sides’ focal adhesions to the nuclear envelope (Figure 3, inset). These fibers, originally named the actin cap by Wirtz et al. [113], are bound by nesprin1/2, which transmits forces generated by the fibers to SUN proteins and up to the nuclear lamina and genome (Figure 3). Thus, this system allows for fast mechanotransduction to the nucleus [60] and is responsible for shaping the nucleus in response to extracellular matrix rigidity [113]. These actin cables act as ropes pushing down and compressing the nucleus [113]. Other perinuclear actin pools and regulators of actin polymerization have been involved in sensing mechanical cues [95,114] and regulating nuclear movement [115], suggesting that multiple pathways controlling perinuclear actin polymerization and contraction are likely to operate upstream of nuclear mechanosensing. Consistent with this idea, the regulation of the nuclear shape and function by the perinuclear actin pools [115] is also modulated by actin binding molecules, such as formin FHOD1 [49] or mechanosensitive filamin A [116].

Actin is also localized to the nucleus where it plays multiple functions, and although its nuclear dynamics play a role downstream of mechanical cues, the underlying mechanisms are still poorly understood [117,118,119]. In keratinocytes, mechanical stretching reduces the amount of nuclear G-actin in an emerin-dependent manner, resulting in gene silencing and less active RNA polymerase II [95], consistent with the role of actin in regulating RNA polymerase II [120]. Thus, the increase in cytosolic filamentous actin observed upon mechanical strain [95,107] could indirectly modulate RNA polymerase II by affecting the balance of cytoplasmic/nuclear G-actin [95]. Multiple actin nucleators and their activators localize also to the nucleus [121], but their nuclear role in response to mechanical strain is unclear. Cytosolic mechanosensitive cytosol actin nucleators such as mDia1 [122] contain nuclear counterparts, such as mDia2, but its role downstream of mechanical forces inside the nucleus is not yet clear [123]. Since mechanical forces regulate formin- and Arp2/3-depedent actin polymerization in the cytosol [114,124,125], it will not be easy to separate these actions from those observed in the nucleus in response to mechanical cues. What is clear is that nuclear actin contributes to preserve nuclear mechanical stability at least under certain conditions [126], but the actin regulators responsible for this are still uncharacterized.

Cytosolic calcium levels play an important role in regulating nuclear and perinuclear actin. These pools of F-actin are dependent on inverted formin-2 (INF2) [17,114,127]. In the cell response to evasion upon cell confinement, calcium accumulation in the nucleus initiates the activation of downstream signaling which results in myosin II-mediated contraction [17,18]. Thus, calcium stores and their interplay with actin polymerization and contraction are important mediators of the cell response to nuclear mechanotransduction.

Cytoplasmic microtubules (MTs) are also important for nuclear shape and heterochromatin regulation [128]. A recent study showed that microtubule acetylation downstream of the pathway sensing substrate rigidity feeds back into RhoA-dependent signaling, resulting in YAP nuclear translocation regulation [129]. Therefore, MTs are also part of the pathways controlling nuclear mechanotransduction.

## 3. Extranuclear Mechanosensors Regulate Nuclear Mechanotransduction

The examples described in the previous section strongly suggest that the nucleus itself has many resources to sense forces, but probably this is not sufficient to cope with the enormous variety of physiological and environmental situations impacting the tensional status of the cell. Thus, other mechanotransduction mechanisms, which arise as a consequence of mechanosensitive cell structures, not necessarily physically connected to the nucleus, also contribute to regulate nuclear response to mechanical inputs. The actin cytoskeleton, in addition to regulate LINC complex mediated nuclear mechanoresponse, also regulates other pathways independently of the LINC complex [130]. Furthermore, actin independent mechanosensitive processes developing at the PM also regulate the nuclear mechanoresponse (see below) (Figure 5).

### 3.1. Actin Cytoskeleton Dependent Regulation of Mechanoresponsive Pathways Targeting Nuclear Biology

A key signaling pathway controlling mechanotransduction pathways is led by the Hippo pathway, which controls organ size and tissue homeostasis by regulating the expression of genes important in cell proliferation, apoptosis and cell differentiation [110]. The Hippo pathway core kinases MST (mammalian STE20-like) and LATS (Large Tumor Suppressor Kinase) regulate the nuclear entry of the transcriptional regulators YAP/TAZ [110]. MST phosphorylate LATS, which in turn phosphorylates YAP/TAZ, inducing its cytoplasmic retention [110]. When MST or LATS are inhibited, dephosphorylated YAP/TAZ are imported into the nucleus [110]. Cell tension controlling pathways upstream of the actin cytoskeleton inhibit MST, or upstream of it, which results in dephosphorylation of YAP/TAZ, its nuclear translocation and activation of its target genes [71,110]. Thus, mechanical cues, through the actin cytoskeleton, activate YAP/TAZ. While the role of the actin cytoskeleton upstream of YAP/TAZ has a clear role [109,135,136], actin independent processes downstream of YAP/TAZ, and its own mechanical plasticity also contribute to increase the YAP/TAZ nucleo-cytoplasmic ratio [42].

Megakaryocytic acute leukemia protein MAL (also known as MRTF-A or MKL1) is a co-activator of serum response factor (SRF), a transcription factor essential for embryogenesis [137]. MAL nucleocytoplasmic shuttling is controlled by its direct binding to G-actin in the cytosol and the nucleus [130,138,139]. G-actin binds MAL and upon actin polymerization and subsequent G-actin reduction, MAL binds the importin α/β complex for its nuclear translocation [138]. In addition, nuclear G-actin facilitates MAL export and blocks MAL activation on SRF [140]. As expected, cell tension and nuclear regulators of the actin cytoskeleton also regulate MAL nuclear localization [141,142,143]. In addition to the paradigmatic examples of mechanoresponsive transcription factors (TFs), including YAP/TAZ and MAL, other TFs are also regulated by mechanical cues, including β-catenin [144], Runx2 [96,145] and others [101,146].

### 3.2. Mechanical Control of Lipid Homeostasis Regulating Pathways

Other mechanoresponsive transcription factors include sterol regulatory element–binding proteins (SREBPs). SREBPs, which include SREBP1 and 2, regulate the expression of proteins dedicated to the synthesis and uptake of cholesterol, fatty acids, triglycerides and phospholipids [147]. Upon lipid starvation, ER localized SREBP moves to the golgi, where it is cleaved [148]. The proteolytic product released into the cytosol translocates to the nucleus to regulate transcription, where it binds and activates genes under the sterol response elements (SREs) [147,148]. Different groups have shown that cell tension controlling pathways regulate SREBP1 transcriptional activity. Integrins, downstream of shear stress, activate SREBP1 [149], while stiff ECM inhibits SREBP1/2 activity through RhoA-dependent pathways [112,150]. Thus, lipid metabolism controlling gene expression is tightly coupled to actomyosin-mediated contraction. Interestingly, SREBP1 interacts with Lamin A/C [151], providing additional ties between mechanosensitive pathways and SREBP1.

### 3.3. Mechanosensitive Plasma Membrane Curvature Regulates Nuclear Function

The Notch signaling pathway functions in cell fate decision and tissue organization [152], and mechanical forces play a major role in regulating this pathway [153] (reviewed in [131]). A unique mechanosensitive process is responsible for Notch activation [131]. Notch signaling starts with the interaction of Notch to its ligands that reside in juxtaposed cells; this binding induces the cleavage of the Notch intracellular domain (NICD), which is imported into the nucleus to regulate transcription [152]. How does this cleavage occur? The binding of the ligand to Notch induces the endocytosis of the ligand, still bound to Notch, which unfolds Notch exposing cryptic binding sites recognized by peptidases, resulting in the cleavage, generation of NICD and its nuclear translocation [131] (Figure 5a). Consistent with these studies, in vitro studies have shown that Notch1 is mechanosensitive [154,155]. This trafficking- and force-driven mechanism [155,156,157,158], although relying also on actomyosin contractility [158,159], and sharing some similarities with intercellular adhesive complexes, is an example of the versatility used by cells to generate forces resulting in unfolding of mechanosensitive domains.

Another example involving endocytic vesicles has been shown for caveolae (Figure 5b) [160]. Caveolae are small PM invaginations (50–80 nm in diameter) that are highly abundant in cells experiencing high mechanical strain [134]. Caveolae are unique in the sense that they flatten out upon high cell tension [14,132,133]. Upon caveolae flattening induced by mechanical strain, the caveolar core components EHD2 and Cavin3 are released from the PM and translocate to the nucleus [161,162]. Nuclear EHD2 regulates gene expression [161,163], while Cavin3 regulates pro-apoptotic and DNA repair pathways [162,164]. These studies suggest that mechanical stress sensed at caveolae initiates mechano-transduction to the nucleus.

## 4. Conclusions

Studies conducted over the last two decades have clearly shown that the main nuclear components are tightly connected to each other, and directly or indirectly to the cytoskeleton. There are in addition multiple ties between the different nuclear components. The cytoskeleton-bound LINC complex contacts the nuclear lamina and the NPC [40,165,166]; in turn the lamina contacts the NPC components [77,167] and the chromatin [103] (Figure 6). These components create a network, whose individual elements are mechanosensitive and their sum creates a supramolecular mechanosensitive structure, which is hard-wired to the extracellular components and juxtaposed cells (Figure 3 and Figure 6). However, in addition to this protein-based flexible architecture, lipid bilayers and their dynamics play a major role (Figure 4 and Figure 5). The NE is mechanosensitive, and like the PM can sense mechanical cues (Figure 4a). In addition, there is a reciprocal interplay between the cytosol/PM and the nucleus. The NE senses mechanical cues and induces a response in the cytosol [17,18,34], whereas the cytosol and the PM sense extranuclear tension and send signaling intermediates to the nucleus [131,134] (Figure 5). This suggests that the cell response to physical cues may be highly coordinated between the different organelles.

Why there are so many mechanisms that sense tension changes at different cell regions that converge into nuclear biology? This is likely to reflect the diversity, in form and magnitude, of the mechanical cues reaching the cell, which require different sensors dispersed throughout the cell, ensuring a global nuclear, and by extension, cell response. In addition, the nucleus not only controls gene expression but also DNA replication, protection, and repair, which is likely to require different pathways downstream of mechanical cues.

Mutations or deletions of many of the genes encoding for the proteins described in this review, derive in serious human pathologies. These pathologies frequently involve malfunctioning of mechanically challenged tissues. Typical pathologies where these pathways are altered include, cardiovascular disease [168], muscular dystrophies [169,170], lipodystrophies [171] and progeria [172]. Thus, it is important to understand the molecular mechanisms by which these proteins function in response to mechanical signals.

## Figures and Tables

**Figure 1 biomolecules-12-00404-f001:**
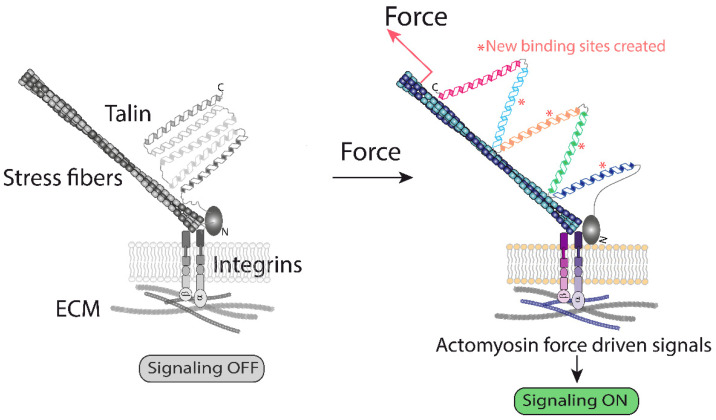
A graphical representation of the ECM–integrin interface generated forces. Activation of integrins leads to mechanical unfolding of the actin cytoskeleton and integrin-bound talin. The current model suggests that activation of actin polymerization downstream of integrins exerts a pulling force on talin, unfolding its cryptic binding sites for vinculin, which starts a signaling cascade [13]. This represents one of the best understood examples of how mechanical forces change protein conformation, triggering biochemical modifications [9,13].

**Figure 2 biomolecules-12-00404-f002:**
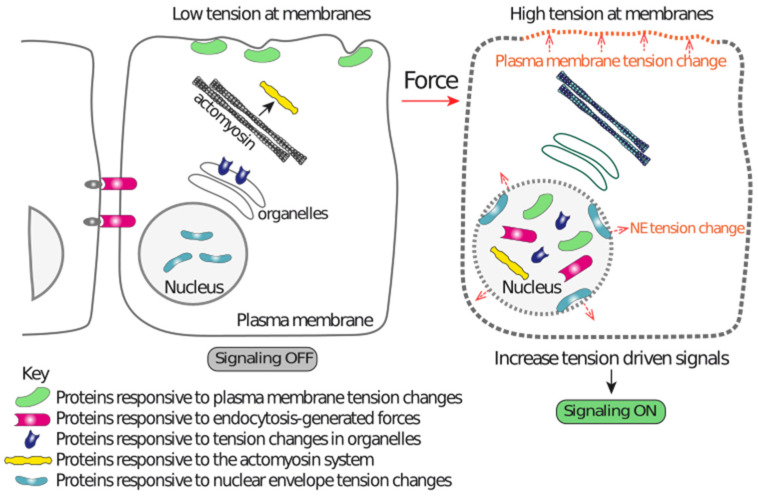
A graphical representation illustrating the different mechanisms acting independently on the ECM-integrin-actin-LINC-lamina-chromatin system that contributes to nuclear mechanoadaptation. Tension alterations in different membranes drive changes in the lipid bilayers that are sensed by proteins, which in turn modify their localization. This localization change drives enrichment in the nucleus or within nuclear regions, resulting in nuclear function regulation. Endocytic events also generate forces that induce downstream events leading to nuclear localization of the pathway effectors. The actomyosin system and forces reaching at organelles also regulate pathways that act independently of the LINC complex. These mechanisms are capable of transducing mechanical information to the nucleus by mechanisms that do not require nuclear-linked multimolecular structures, such as those shown in Figure 3 (ECM-integrin-actin cytoskeleton-LINC-lamina-chromatin).

**Figure 3 biomolecules-12-00404-f003:**
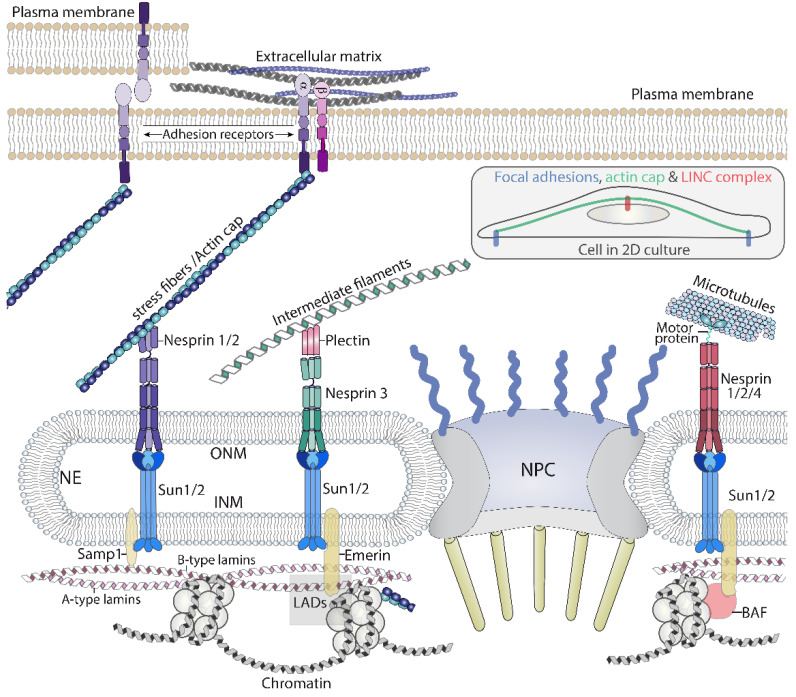
A graphical representation of the main components of the chain of macromolecular complexes that transmit forces, initiated outside the cell, to the nucleus. The LINC complex, through nesprins, connects with the three cytoskeletons, which permits the nucleus to sense forces generated or sensed by these polymers. LINC: linker of nucleoskeleton and cytoskeleton. NPC: Nuclear pore complex. ONM: Outer nuclear membrane. INM: inner nuclear membrane. LAD: Lamin associated domains. The inset illustrates the actin cap bound to focal adhesions and the LINC complex while crossing over the nucleus.

**Figure 4 biomolecules-12-00404-f004:**
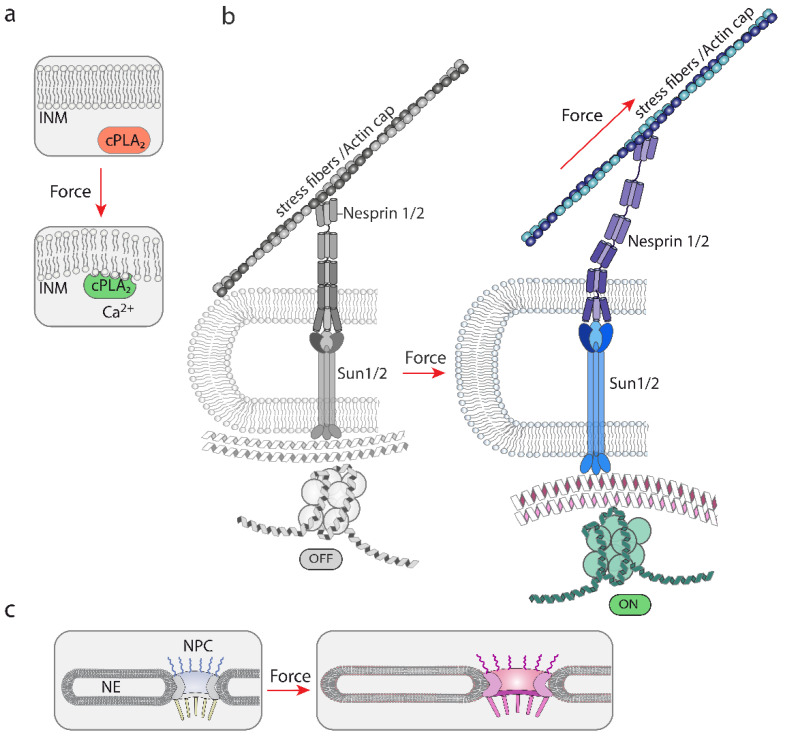
A graphical representation of the different proposed mechanisms of mechanosensing by nuclear structures. (**a**) The NE is mechanosensitive. Tension changes modify the structure of the NE, which modifies the binding sites for the mechanosensitive cPLA_2_. Calcium accumulation contributes to the activation of cPLA_2_ [34]. Forces that alter lipid bilayers can be generated by osmolarity variations leading to changes in osmotic pressure. (**b**) A graphical representation of the model by which unfolding events in nesprin and SUN may lead to modifications in lamins and chromatin. This system is believed to rely on mechanical forces generated by actin polymerization and/or contraction of the actomyosin complex, which would pull from associated proteins, such as nesprins. These actions generate forces that would transmit mechanical cues up to the chromatin [2,40]. (**c**) A graphical representation illustrating the proposed sensitivity and adaptation of the NPC diameter to mechanical cues [41,42,43]. Hyperosmolarity [43] and ECM rigidity [42] have been shown to alter the NPC diameter.

**Figure 5 biomolecules-12-00404-f005:**
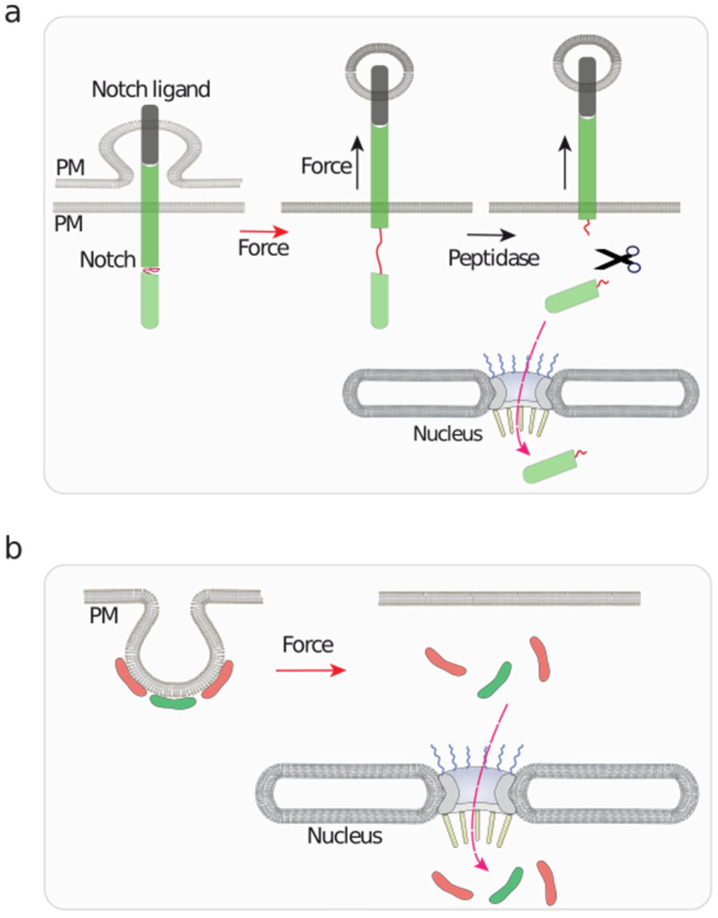
A graphical representation of mechanosensitive events taking place at the plasma membrane with implications in nuclear functions. (**a**) Notch binding to its ligand initiates an endocytic process of the ligand, which exerts a pulling force on bound Notch. This stretching action unfolds Notch, unmasking cryptic peptidase sensitive sites, which triggers the release of a Notch fragment that is translocated to the nucleus to regulate gene expression [131]. (**b**) Caveolae are mechanosensitive plasma membrane (PM) invaginations that undergo flattening upon osmotic swelling [14,132,133]. This triggers the release, and subsequent nuclear translocation of EHD2 and Cavin3, represented as red and green modules, respectively, which regulate nuclear function [134].

**Figure 6 biomolecules-12-00404-f006:**
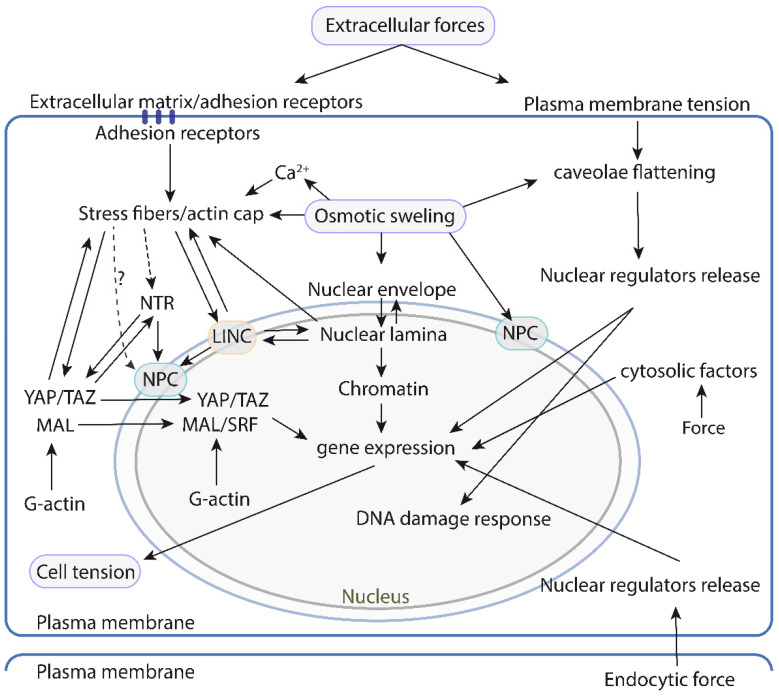
A map of the physical and/or functional interactions between the different mechanosensitive or mechanoresponsive cell structures and pathways controlling nuclear mechanotransduction and function. The left side focuses on the pathways that rely on the actin cytoskeleton and the LINC complex to transduce mechanical information to the nucleus. This system is relatively well understood, and the current understanding suggests that it is highly interconnected with different mechanosensitive structures throughout. The right side illustrates the pathways that operate independently of the actin cytoskeleton-LINC axis and respond to forces generated or sensed at the plasma membrane. Mechanoresponsive events in cytosolic organelles that transduce to the nucleus have also been described and are depicted on the right side. LINC: linker of nucleoskeleton and cytoskeleton. NPC: Nuclear pore complex. NTR: nuclear transport receptors.

## Data Availability

Not applicable.

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
