# Peer review of "A Multisensory Network Drives Nuclear Mechanoadaptation"

_biomolecules, 2022, doi:10.3390/biom12030404_

Round 1
Reviewer 1 Report
In this manuscript entitled “A multisensory network drives nuclear mechanoadaptation”, the author presents a comprehensive and compelling review on the molecular mechanisms underlying the transduction of mechanical forces into the nucleus and its elicited biochemical and structural responses. The overall text is well written, the logic flow is clear and makes for an enjoyable and informative read. I think that in its present form is ready for publication, but I would like to ask the author to address some minor things before the manuscript is ready for prime time.
Minor comments
- For the sake of clarity and to help the reader I would suggest the author to edit Figure 2. As it stands, it is not straight forward to interpret that the membrane is subjected to some kind of tension, apart from the label indicating so. Also, the change in color, shape and localization of the cartoons identifying the different proteins are a bit confusing, as it is not clear who is who and what kind of change in localization they are going through.
- Line #28: In this sentence: “Adaptation of cells to environmental changes is essential for cell survival, successful proliferation and homeostasis.” I would suggest the author to change the order to “homeostasis, cell survival and successful proliferation.”
- Lines #109 and 112, there are two contiguous sentences starting with “Therefore …”, so please, double check and correct the style throughout the text.
- Line #222: Correct “This approached showed” with “This approach showed”.
- In line #252 the author suggests that the NPC architecture is altered in response to mechanical forces, but this might not be completely accurate and potentially confusing. The latest papers from Zimmerli et al. 2021 and Akey et al. 2022 show that the overall NPC architecture is conserved upon constriction/expansion. It appears that localized conformational changes of certain components occur with (citing from Zimmerli et al) “an overall preserved intrasubcomplex arrangement” and that increased separation of the NPC protomers is the main molecular mechanism driving the expansion/constriction transition leading to changes in diameter of the central channel.
- An interesting review on lamina architecture and their role in mechanotransduction was (very) recently published by Vahabikashi et al. (DOI 10.1063/5.0082656), I would suggest the author to include it in section 2.4, just in case a reader would like to explore that specific topic more in depth.
- Line #356: Please correct this sentence “A direct mechanist to transduced mechanical forces sensed at focal adhesions is mediated […]”.
Author Response
In this manuscript entitled “A multisensory network drives nuclear mechanoadaptation”, the author presents a comprehensive and compelling review on the molecular mechanisms underlying the transduction of mechanical forces into the nucleus and its elicited biochemical and structural responses. The overall text is well written, the logic flow is clear and makes for an enjoyable and informative read. I think that in its present form is ready for publication, but I would like to ask the author to address some minor things before the manuscript is ready for prime time.
I want to thank the reviewer for her/his comments and input. Her/his analysis allowed me to improve the review and correct some mistakes.
Minor comments
- For the sake of clarity and to help the reader I would suggest the author to edit Figure 2. As it stands, it is not straight forward to interpret that the membrane is subjected to some kind of tension, apart from the label indicating so. Also, the change in color, shape and localization of the cartoons identifying the different proteins are a bit confusing, as it is not clear who is who and what kind of change in localization they are going through.
- I agree that this figure was too general and vague. I have made some changes to improve the message.
- Line #28: In this sentence: “Adaptation of cells to environmental changes is essential for cell survival, successful proliferation and homeostasis.” I would suggest the author to change the order to “homeostasis, cell survival and successful proliferation.”
- Makes a lot of sense, thanks!
- Lines #109 and 112, there are two contiguous sentences starting with “Therefore …”, so please, double check and correct the style throughout the text.
- These and other sections were modified as suggested.
- Line #222: Correct “This approached showed” with “This approach showed”.
- Sorry for this mistake, now corrected.
- In line #252 the author suggests that the NPC architecture is altered in response to mechanical forces, but this might not be completely accurate and potentially confusing. The latest papers from Zimmerli et al. 2021 and Akey et al. 2022 show that the overall NPC architecture is conserved upon constriction/expansion. It appears that localized conformational changes of certain components occur with (citing from Zimmerli et al) “an overall preserved intrasubcomplex arrangement” and that increased separation of the NPC protomers is the main molecular mechanism driving the expansion/constriction transition leading to changes in diameter of the central channel.
- Thanks for pointing this out; I apologize for not making the right interpretation of these studies. The sentence is now modified, and the term “architecture” is substituted with “diameter”. The reference of Akey et al. is now included in this section.
- An interesting review on lamina architecture and their role in mechanotransduction was (very) recently published by Vahabikashi et al. (DOI 10.1063/5.0082656), I would suggest the author to include it in section 2.4, just in case a reader would like to explore that specific topic more in depth.
- I have included this reference in section 2.4 for readers wanting to further explore this topic.
- Line #356: Please correct this sentence “A direct mechanist to transduced mechanical forces sensed at focal adhesions is mediated […]”.
- Sorry for this mistake, now it is corrected.
Reviewer 2 Report
Overall, this is a very well organized updated review of a lot of important findings. The article itself is delightful.
Some minor points
- Add description to the modules used in fig.2 will help the readers more.
- Line 137, The nuclear envelope is key => “the key”
- In Figure 4 legend, it would be good to add some description about “Force”. Maybe give some examples.
- Line 347, this sentence is too fragmented. “The actin cytoskeleton and myosin II complex, known as the actomyosin system, forms, with the aid of other actin binding proteins, the stress fibers”
- Line 356, “A direct mechanist” does it mean “mechanism”?
- 5, please describe what are green and red modules in the legends.
- Line 451, “where is cleaved” -> where it is cleaved.
- Fig. 6 provides a very good illustrative summary of this review. If more description can be added to the legend may help readers to organize all information.
Author Response
Overall, this is a very well organized updated review of a lot of important findings. The article itself is delightful.
I want to thank the reviewer for her/his comments and input. Her/his analysis allowed me to improve the review and correct some mistakes.
Some minor points
- Add description to the modules used in fig.2 will help the readers more.
This figure was a bit vague. I have made some changes hoping to solve the problem.
- Line 137, The nuclear envelope is key => “the key”
This sentence has been modified, thanks.
- In Figure 4 legend, it would be good to add some description about “Force”. Maybe give some examples.
Forces acting on the different nuclear mechanosensitive systems are described now.
- Line 347, this sentence is too fragmented. “The actin cytoskeleton and myosin II complex, known as the actomyosin system, forms, with the aid of other actin binding proteins, the stress fibers”
I agree. The sentence is now divided into two.
- Line 356, “A direct mechanist” does it mean “mechanism”?
Yes, now corrected.
- 5, please describe what are green and red modules in the legends.
Done.
- Line 451, “where is cleaved” -> where it is cleaved.
This has been corrected now.
- Fig. 6 provides a very good illustrative summary of this review. If more description can be added to the legend may help readers to organize all information.
I have now added more information; hopefully now it is easier to follow the figure.